# Physiology of gamma-aminobutyric acid treated *Capsicum annuum* L. (Sweet pepper) under induced drought stress

Babar Iqbal[1]*, Fida Hussain[1,2], Muhammad Saleem khan[2], Taimur Iqbal[3], Wadood Shah[4], Baber Ali[5], Khalid M. Al Syaad[6], Sezai Ercisli[7,8]

1 Department of Chemical & Life Sciences, Qurtuba University of Science and Information Technology, Peshawar, Pakistan, 2 Department of Botany, Islamia College Peshawar, Peshawar, Pakistan, 3 Faculty of Crop Protection Sciences, Department of Plant Pathology, University of Agriculture, Peshawar, Pakistan, 4 Biological Sciences Research Division, Pakistan Forest Institute, Peshawar, Pakistan, 5 Department of Plant Sciences, Quaid-i-Aazam University, Islamabad, Pakistan, 6 Department of Biology, College of Science, King Khalid University, Abha, Saudi Arabia, 7 Faculty of Agriculture, Department of Horticulture, Ataturk University, Erzurum, Türkiye, 8 HGF Agro, Ata Teknokent, Erzurum, Türkiye

* Babariqbal441@gmail.com

**Data Availability Statement:** All relevant data are within the manuscript.

**Funding:** This study is funded by Deanship of Scientific Research, King Khalid University. Award Number: R.G.P2/345/44.

## Abstract

There is now widespread agreement that global warming is the source of climate variability and is a global danger that poses a significant challenge for the 21st century. Climate crisis has exacerbated water deficit stress and restricts plant's growth and output by limiting nutrient absorption and raising osmotic strains. Worldwide, Sweet pepper is among the most important vegetable crops due to its medicinal and nutritional benefits. Drought stress poses negative impacts on sweet pepper (*Capsicum annuum* L.) growth and production. Although, γ aminobutyric acid (GABA) being an endogenous signaling molecule and metabolite has high physio-molecular activity in plant's cells and could induce tolerance to water stress regimes, but little is known about its influence on sweet pepper development when applied exogenously. The current study sought to comprehend the effects of foliar GABA application on vegetative development, as well as physiological and biochemical constituents of *Capsicum annuum* L. A Field experiment was carried out during the 2021 pepper growing season and GABA (0, 2, and 4mM) concentrated solutions were sprayed on two *Capsicum annuum* L. genotypes including Scope F1 and Mercury, under drought stress of 50% and 30% field capacity. Results of the study showed that exogenous GABA supplementation significantly improved vegetative growth attributes such as, shoot and root length, fresh and dry weight, as well as root shoot ratio (RSR), and relative water content (RWC) while decreasing electrolyte leakage (EL). Furthermore, a positive and significant effect on chlorophyll a, b, a/b ratio and total chlorophyll content (TCC), carotenoids content (CC), soluble protein content (SPC), soluble sugars content (SSC), total proline content (TPC), catalase (CAT), and ascorbate peroxidase (APX) activity was observed. The application of GABA at 2mM yielded the highest values for these variables. In both genotypes, peroxidase (POD) and superoxide dismutase (SOD) content increased with growing activity of those antioxidant enzymes in treated plants compared to non-treated plants. In comparison with the rest of GABA treatments, 2mM GABA solution had the highest improvement in morphological

**Competing interests:** The authors have declared that no competing interests exist.

**Abbreviations:** APX, Ascorbate peroxidase; CAT, Catalase; CC, Carotenoids content; Chl "a", Chlorophyll "a"; Chl "b", Chlorophyll "b"; Chl a/b, Chlorophyll a/b ratio; EL, Electrolyte leakage; MDA, Malondialdehyde; POD, Peroxidase; RDW, Root dry weight; RFW, Root fresh weight; RL, Root length; ROS, Reactive oxygen species; RSR, Root shoot ratio; RWC, relative water content; SDW, Shoot dry weight; SFW, Shoot fresh weight; SL, Shoot length; SOD, Superoxide dismutase; SPC, Soluble protein content; SSC, Soluble sugar content; TCC, Total chlorophyll content; TPC, Total proline content.

traits, and biochemical composition. In conclusion, GABA application can improve development and productivity of *Capsicum annuum* L. under drought stress regimes. In addition, foliar applied GABA ameliorated the levels of osmolytes and the activities of antioxidant enzymes involved in defense mechanism.

## 1. Introduction

Climate change has affected the world in many ways, including threatening its plant life [1]. Many adverse situations are continuously being faced by plants in both agricultural as well as natural environments. These abrupt changes in climate cause various biotic and abiotic strains and such small variations in climatic condition can prompt the expressions or repressions of several genes in plants, resulting in decline in crop production [2,3]. Many research studies have shown that rise in temperature due to climatic variability is likely to affect many agricultural crops such as *Triticum aestivum* L., *Capsicum annuum* L., *Zea mays*, and *Oryza sativa* from 3% to 10% per degree of warming above historic level [4]. Drought stress is one of the topmost ecological factors that adversely affect the development, production and performance of the plants exclusively during the vegetative and growth stage by reducing the crop yield produced for economical purposes or existence [5,6].

Areas throughout the world that are half or entirely dry are more prone to water scarcity and drought stress, and would probably face increased water limited conditions as a result of climate change and global warming, particularly during the summer seasons in many parts of the world [7]. To fulfill the food demand of fastly growing global population, which is predicted to reach up to 9.8 billion by 2050, global agricultural production must be quadrupled. Conversely, natural resources, particularly land and water, are rapidly depleting and eroding. Historical data and model simulations showed a significant danger of drought around the globe [8]. According to FAO (Food and Agriculture Organization) report up to 26% arable land of the total biosphere is distressed by water deficiency stress [9].

Pakistan comes in the arid and semiarid zones which directly makes it vulnerable to water stress problems similar to any other emerging nations in this part of the globe. Pakistan's 1/4th (4.9 million hectare) land is prone to drought and the situation is getting worse by each day passed [10]. All the provinces of Pakistan are facing major water deficient conditions especially the major parts of Balochistan and Sindh [11]. According to an estimated rise in demand by 2025, Pakistan would require 274 million Acre Feet of water supply to suit its agricultural productivity demands. Previous research studies indicate that the availability of water by 2025 is unlikely to vary from the current 191 MAF availability, the difference might widen [12]. Pakistan breached the water shortage threshold in 2005, and if the current trend continues, the country will probably reach the extreme water scarcity threshold by 2025 [13].

To cope with drought, plants employ a variety of stress-coping mechanisms, including the activation of antioxidant enzymatic defense mechanisms, generation of stress hormones such as ethylene and abscisic acid, and the modifications of shoot and root architecture [14,15]. Drought stress reduces the rate of photosynthesis and respiration, translocation, ion uptake, content of chlorophyll, carbohydrates, nutrient metabolism, and various growth hormones [16]. Water deficiency stress disturbs the equilibrium between reactive oxygen species (ROS) production and the antioxidant defense system which cause the accretion of ROS by inducing oxidative stress to protein, lipids membrane and DNA strands disruption [17].

Exogenous administration of plant growth regulators, such as glutamic (Glu), proline (Pro), and polyamines, and γ-aminobutyric acid has recently been identified as an effective approach

for alleviating drought damage in plants [18]. GABA is a non-protein amino acid with 4-carbon structure, it has the ability to provide multiple protections to plants against several abiotic stresses under induce tolerance such as stimulation, and mechanical injury, hormonal toxicity, extreme temperatures, salinity stress, waterlogging and water-deficient stress [19,20]. GABA has been demonstrated to have numerous roles in plant species, including control of proline and tricarboxylic acid (TCA) cycle metabolism, maintenance of carbon, oxidative damage mitigation and nitrogen metabolism linked with tolerance to abiotic stressors [21]. In the regulation of plants growth and tolerance against various abiotic stresses GABA is numerously involved in countless physiological and biochemical functions such as C–N fux for tricarboxylic acid cycle, carbon/nitrogen metabolism regulation of redox status and osmotic pressure [22]. All these reports against a number of abiotic stresses in plant species treated with GABA suggested that there is potential means of enhancing abiotic stress tolerance in plants.

*Capsicum annuum* L. belonging to family Solanaceae, has a number of medicinal benefits and economically important crop [23]. It is considered the 3rd greatest monetary gain crop of family Solanaceae right after potato and tomato [4]. This plant is indigenous to North and South America. Sweet pepper is considered as the cash crop and some of the supreme valuable crops in many countries including Pakistan where its share is 1.5% in the GDP and is the sixth highest sweet pepper exporter all over the world [24]. Sweet pepper is extensively cultivated in Pakistan on 157.9 thousand hectares producing 142.9 thousand tons with a regular yield of 3.3 tons/hec [25]. *Capicum annuum* L. is mostly grown in rain fed regions throughout the world and mostly undergo water deficient conditions in the summer season of the year due to less rainfall caused by climate change. Those areas which are already water deficient the total production of sweet pepper is grimly affected by drought which at the same time may bring changes in the plant's characteristics such as physio-biochemical, morphological, and metabolic modification in many plant's organs [26].

The aim of the current research study was to evaluate the biochemical, physiological, and growth features of *Capsicum annuum* L. (Scope F1 and Mercury) under various exogenous GABA application levels, as well as to identify its potential efficacy in countering the negative effects of water deficit stress. Moreover, to figure out how adaptable is GABA in controlling crucial metabolic processes by improving the drought tolerance capacity of *Capsicum annuum* L. exposed to various levels of drought induced stress regimes.

## 2. Materials and methods

### 2.1. Site description and experimental design

Field trials were carried out in natural environment at the Department of Chemical & Life Sciences, Qurtuba University of Science and Information Technology Peshawar, Pakistan (33˚ 58' 49.0" N and 71˚ 26' 45.9" E) during the pepper growing season of 2021. Climate of Peshawar is tropical and temperature ranges from 5˚ degrees Celsius (in January) to 39˚ degrees Celsius (in June). The seeds of two *Capsicum annuum* L. varieties, Mercury and Scope F1 were supplied by Pakistan's National Agriculture Research Center (NARC). The *Capsicum annuum* L. Surface sterilized seeds were planted in clay containers (height 20 cm, lower and upper diameter 18cm, while thickness is 2cm), which were filled with 2kg of soil in a 3:1 ratio (sand, loam) with well-composted farmyard manure. Pots were housed in a nursery with a 3×1.5 m$^2$ space and were positioned 5 cm apart in a full randomized block design (RCD) to allow for appropriate air flow and daylight exposure. Watering was done on regular basis and weeds were removed by hand.

The seedlings were divided/thinned into three per replicate after 15 days of germination. After the water deficiency treatments were correctly established, the GABA treatment began at

the fourth leaf stage. To boost leaf adhesion, a little amount of polyoxyethylenesorbitan mono-laurate (Tween-20) was added as a surfactant to a GABA solution. Aqueous GABA Solution of 0, 2, and 4mM sprayed onto both sides of the leaves in such a way that it thoroughly gets soaked. All the plants were subjected to 3 stages of water scarcity stress until trials (which lasted about four months): stressed free circumstances (the control group i.e., complete irrigation), moderate stress (50% of field capacity), and severe stress (30% of field capacity). 54 *Capsicum annumm* L. plants were collected at 80% maturity in this experiment (in the green stage). The physiological parameters were evaluated in the top portion of the seedlings (leaf), and the samples were rapidly iced up in liquid nitrogen after harvest and stored at -80˚C for examination of growth components, osmoprotectants, and plant antioxidant enzymes, as well as germination parameters. Throughout the trial, all normal practices for healthy and disease and pest-free crops were followed on a regular basis.

## 2.2. Soil analysis

Prior to planting, soil sample (0 to 30 cm) was taken from the experimental location for soil physio-chemical investigation. The soil's chemical properties were as follows: pH = 6.2, electrical conductivity (EC) = 0.383 ds/m [27], % field capacity 8.69–31.02% [28], organic carbon (C) content was 0.55% [29], nitrogen (N) level was 0.027 g/kg [30], and accessible phosphorus (P) level was 0.053 g/kg [31] and available potassium (K) was 0.014 g/ kg [32]. The soil textural class was silt loamy as calculated by the hydrometer method [33].

## 2.3. Agronomic characteristics

**2.3.1. Growth parameters.** The fresh weight of pepper seedling shoots in each replication was measured immediately after harvest using a digital scale. The dry weight of the shoots was recorded after oven drying at 70 ˚C until stable mass was attained. A meter rod was used to measure the length of the sweet pepper roots and shoots. The sweet pepper plant roots were gently taken from the containers, and after multiple washes with water, the fresh weights were measured using a digital scale. They were then roasted at 70 ˚C to ascertain the dry weights of the roots of sweet pepper plant [34].

**2.3.2. Root shoot ratio (RSR).** Method of [35] was employed to conclude this parameter. The dry mass (g) of both the shoot and root was measured after being kept in an oven for 72 hours for further calculations.

$$RSR = \frac{root\,dry\,mass}{shoot\,dry\,mass} \tag{1}$$

**2.3.3. Relative water content (RWC).** An approach described by [36] was used to estimate the relative water content (RWC) of leaves. During this process, fully developed leaves were selected and their fresh weights were recorded. These leaves were then placed in test tubes with distil water at room temperature for around 3 hours. Their turgid weights were measured. After drying for 48 hours at 70 ˚C, the dry weight of the leaves was calculated. The relative water content of the leaf was calculated using the equation below:

$$RWC\,(\%) = FW - \frac{DW}{TW} - DW \times 100 \tag{2}$$

**2.3.4. Electrolyte leakage (EL).** The electrolyte leakage of *Capsicum annuum* L. leaves were estimated conferring to method of [37]. Garden-fresh leaves were sliced into 0.1-g discs.

To remove residues and contaminants from the sample's, distilled water was administered. After that, the leaf samples were put in 10-ml test tubes filled with distilled water. The tubes were incubated at 30 ˚C for 30 minutes before measuring the solution's primary electric conductivity (EC1). While measuring secondary electric conductivity (EC2), the tubes were submerged in a 100 ˚C water bath for 15 minutes. Finally, the overall EC was computed by the following equation:

$$EL\ (\%) = \left(\frac{EC1}{EC2}\right) \times 100 \tag{3}$$

## 2.4. Investigation of plant's physiological and biochemical components

**2.4.1. Estimation of photosynthetic pigments (chlorophyll a, b and carotenoids).** The chlorophyll content of the leaves was assessed using the [38] technique. Fresh pepper plant leaves (0.5 gm) were diced up in 80% 10ml acetone with a pestle and mortar and centrifuged for five minutes and were kept in incubation for 24 hours in the dark at 4 ˚C. Chlorophyll content was then determined using supernatant. Absorbance at 649 nm, 663 nm, and 470nm for chlorophyll "a", chlorophyll "b", and carotenoids was measured using a spectrophotometer against an 80% acetone blank.

**2.4.2. Estimation of Soluble sugar content (SSC).** For the determination of soluble sugar content, the usual [39] methodology, with minor changes. The pepper plant's 0.5-gram foliar material was crushed and then homogenized in 10ml distilled water before adding 1ml of 30% phenol to 0.1ml supernatant and incubating the trials for 20 minutes. After incubation, 5.0ml of rigorous $H_2SO_4$ was supplemented, and values at 420nm were taken.

**2.4.3. Estimation of Total proline content (TPC).** For the approximation of proline content, [40] method was used by homogenizing fresh leaves (0.5-gram) in 10ml of 3% (w/v) sulfosalicylic acid and then filtering the suspension. In a test tube, take 2ml filtrate, and add 2ml glacial acetic acid, 2ml acid ninhydrin (1.87g ninhydrin + 40ml glacial acetic acid + 30ml of 6M phosphoric acid) and warm it at 100 ˚C for 1 hour. The entire process was halted in an ice bath. 4ml of Toluene was used for extraction, and absorbance at 520 nm was measured.

**2.4.4. Estimation of soluble protein content (SPC).** The SPC of the leaf was computed by the methodology of [41] via BSA as standard.

1. 1 gram of monosodium phosphate (MSP) was liquefied in distil water.

2. 9.11 gram disodium phosphate (DSP) liquefied in 170 ml of distil water.

To achieve the desired pH of 7.5 for the phosphate buffer, monosodium phosphate of 32ml and disodium phosphate of 168ml were mixed. Reagent A was prepared by dissolving 3 gram of sodium carbonate ($Na_2CO_3$), 1.5 gram of Na-k tartrate and 0.6 gram of Sodium hydroxide (0.1 N) in 150 ml of distil water. For preparation of Reagent B $CuSO_4.5H_2O$ (0.125 gram) was dissolved in 25ml of distil water. Reagent C was created by combining solution a (150ml) and solution b (3ml). Reagent D: 10ml of foline phenol reagent was diluted with 10ml of distil water in a 1:1 ratio.

In phosphate buffer solution of 1ml of pH 7.0, foliar material of 0.5-gram were grounded and homogenized for 10 minutes. A total volume of 1ml was obtained by adding distil water to 0.1 ml of extract and mixing it with 1ml of reagent having [0.75g sodium carbonate ($Na_2CO_3$), 0.1 N sodium hydroxide (NaOH), and 0.37g sodium potassium tartrate (Na-K tartrate) in distilled water of 40ml]. Shake the solution for ten minutes before adding 0.1ml foline phenol reagent to a test tube and incubating for thirty minutes. At 650 nm, the absorbance of the sample was noted via a spectrophotometer.

**2.4.5. Estimation of peroxidase activity (POD).** The methodology of [42] was employed for peroxidase activity. The plant's fresh leaves of 0.5-gram were homogenized and centrifuged for 20 minutes with a 2ml solution containing [0.2ml phosphate buffer solution (pH 7.0), 12.5g (PVP) and 4.6g ethylene diamine tetra acetic acid (EDTA) in distilled water of 125ml]. The reaction mixture (3ml) included 0.1ml supernatant, 1.3ml methyl ethyl sulfonic acid (MES) buffer (970 mg MES in 50ml distil water), 0.1ml phenyl diamine (36mg in 4ml distil water), and a droplet of 0.3% $H_2O_2$. A spectrophotometer was used to record the changes in absorbance for 3 minutes at 485nm.

**2.4.6. Estimation of superoxide dismutase activity (SOD).** The typical approach of [43] was used for the measurement of SOD concentration in the leaves. The reaction mixture of 3ml containing supernatant of 0.1ml, 0.72 ml methionine (58mg methionine dissolved in 30ml distil water), nitroblue tetrazolium (NBT) of 0.72ml (1.89mg nitroblue tetrazolium dissolved in 30ml distil water), 0.72ml Ethylenediaminetetraacetic acid (1.1mg Ethylenediaminetetraacetic acid dissolved in 30ml distilled water) and riboflavin of 0.72ml (0.02mg riboflavin dissolved in 30ml distilled water) after that the mixture was incubated in the dark and then in the light for approximately a period of 30 minutes and spectrophotometer was used for taking measurements at 560nm.

**2.4.7. Estimation of catalase activity (CAT).** Catalase content (CAT) in foliar material of *Capsicum annuum* L. plant was determined using the approved approach of [44] with certain modifications. The reaction mixture (1ml) was consisted of 0.4ml of 100mM potassium phosphate buffer (pH 7.0), 0.4ml of 30% $H_2O_2$, and 0.2ml of enzyme extract. The reduction in absorbance value at 240nm after 3 minutes of $H_2O_2$ breakdown was observed.

**2.4.8. Estimation of ascorbate peroxidase activity (APX).** Content of APX was measured by the recommended procedure of [45]. The reaction mixture of 1ml was made by homogenizing fresh foliar material (0.5g) with 5ml buffer solution and centrifuging it at least for ten minutes. Enzyme extract of 0.1 ml was combined with 1.8ml of 50mM phosphate buffer with a pH of 7.0, and 0.1ml of 0.5mM ascorbic acid, and 1mL of 30% $H_2O_2$. The absorbance was measured as a decrease occurred at 290 nm for three minutes when compared to a blank 30% $H_2O_2$.

## 2.5. Statistical analysis

For the statistical analysis, a factorial design with varying levels of induced drought stress was utilized. Data for numerous germination characteristics, physiological and biochemical components were calculated using Statistix 10 and SPSS Statistics 25 (SPSS Inc, Chicago IL). These software's were used to perform two-way anova, principal component analysis, regression and correlation, and the Least Significant Differences (LSD) tests. Excel software was also used to create bar graphs. Significance Difference at *P* values, as well as mean separation and standard deviation, were employed to establish significance.

## 3. Results

### 3.1. Effect of GABA on root length, root fresh weight, root dry weight, relative water content and electrolyte leakage under induced drought stress

Results (Fig 1a–1c) indicated that induced drought stress caused a pronounced decline in growth parameters. The application of GABA foliar spray resulted in maximum increase of roots length (RL), root fresh weight (RFW) and root dry weight (RDW) under 50% and 30% field capacity drought stress in both the genotypes, including Scope F1 and Mercury. The use of 2mM GABA treatment led to maximum growth value of RL, RFW and RDW under 50% and 30% field capacity drought stress in both varieties.

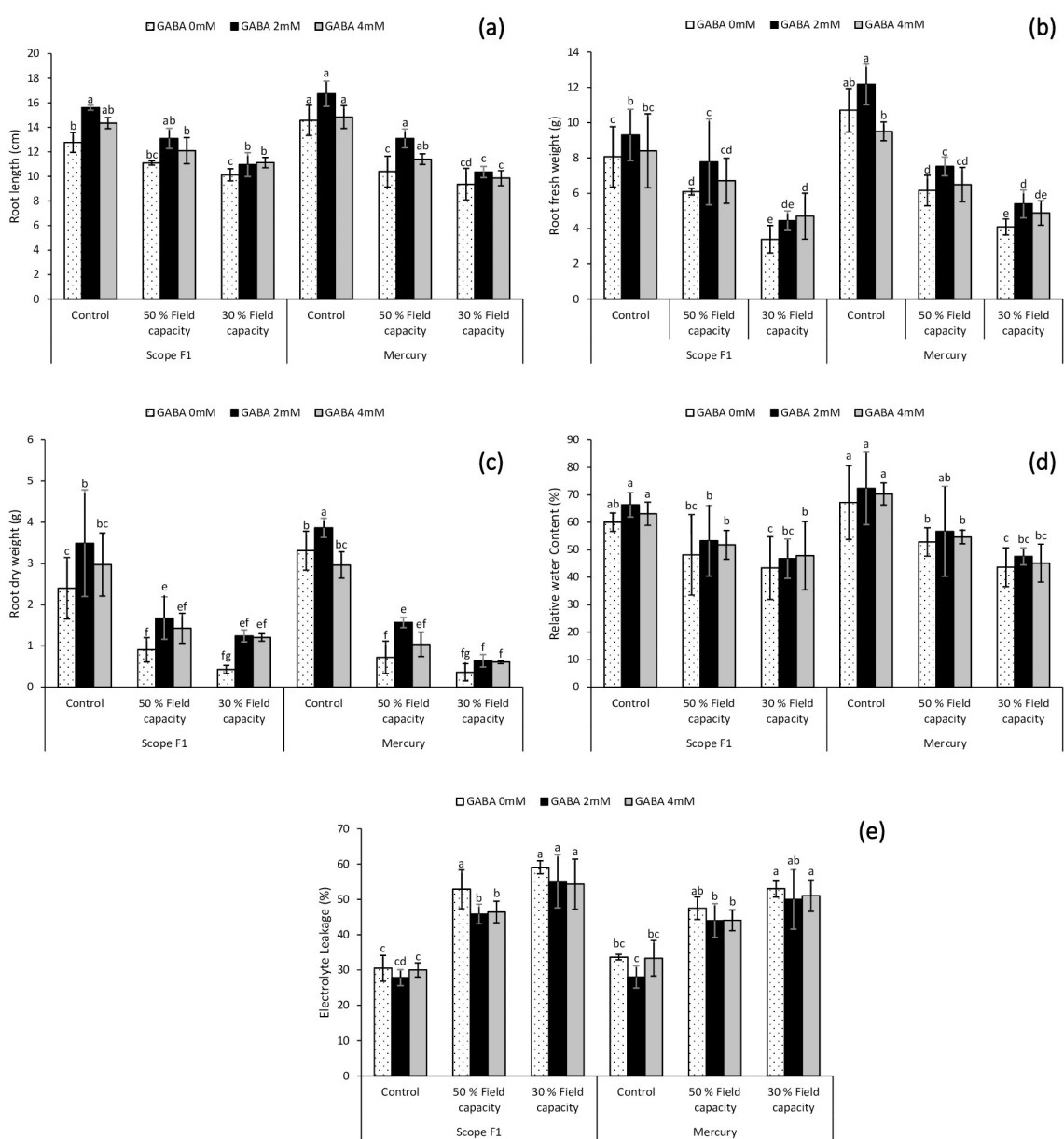

**Fig 1.** Effect of varying levels of exogenously applied GABA on root length (**a**) root fresh weight (**b**) root dry weight (**c**) relative water content (**d**) electrolyte leakage (**e**) of sweet pepper (*Capsicum annuum* L.) grown under varying drought stress conditions. Letters (A–G) indicating least significance difference among the mean values at $p \leq 0.05$.

The *Capsiucm annuum* L. varieties grown under drought stress conditions of 50% and 30% field capacity resulted in higher levels of EL and lower levels of RWC as compared to those grown under 100% field capacity (Fig 1d and 1e). GABA spray significantly enhanced both RWC and RL levels of the *Capsicum annuum* L. plants. In Mercury variety RWC was higher in 50% and 30% field capacity drought stress at 2mM GABA. However, Scope F1 variety results indicated that RWC was higher in 50% field capacity drought stress treated with 2mM GABA whereas in 30% field capacity drought RWC was slightly higher with 4mM GABA treatment as compared to 2mM GABA application. The application of GABA foliar spray evidently reduced

the EL in stressed *Capsicum annuum* L. plants. EL lowered values were reported in 50% and 30% field capacity drought stress in both the varieties treated with 2mM and 4mM GABA application. This analysis of the interplay between drought stress and GABA foliar spray amount revealed that when drought stress worsens, so does EL takes place. In the contrary, Using GABA under water deficit stress condition inhibits EL.

## 3.2. Effect of GABA application on shoot length, shoot fresh weight, shoot dry weight and root shoot ratio under induced drought stress

According to the results, drought stress significantly affected the growth attributes of sweet pepper plant varieties. Exogenous application of GABA spray resulted in maximum values of shoot length (SL), shoot fresh weight (SFW), shoot dry weight (SDW) and RSR under 50 percent and 30 percent field capacities induced drought stress in both the varieties of *Capsicum annuum* L., including Scope F1 and Mercury. In addition, the administration of 2mM GABA resulted in the highest value of SL, SFW and SDW under both the induced drought stress conditions, irrespective of the cultivars. Similarly, maximum value of RSR was reported in 50% and 30% field capacity drought stress in both the varieties treated with GABA. Exogenous treatment of GABA solution considerably improved growth attributes of *Capsicum annuum* L (Fig 2a–2d).

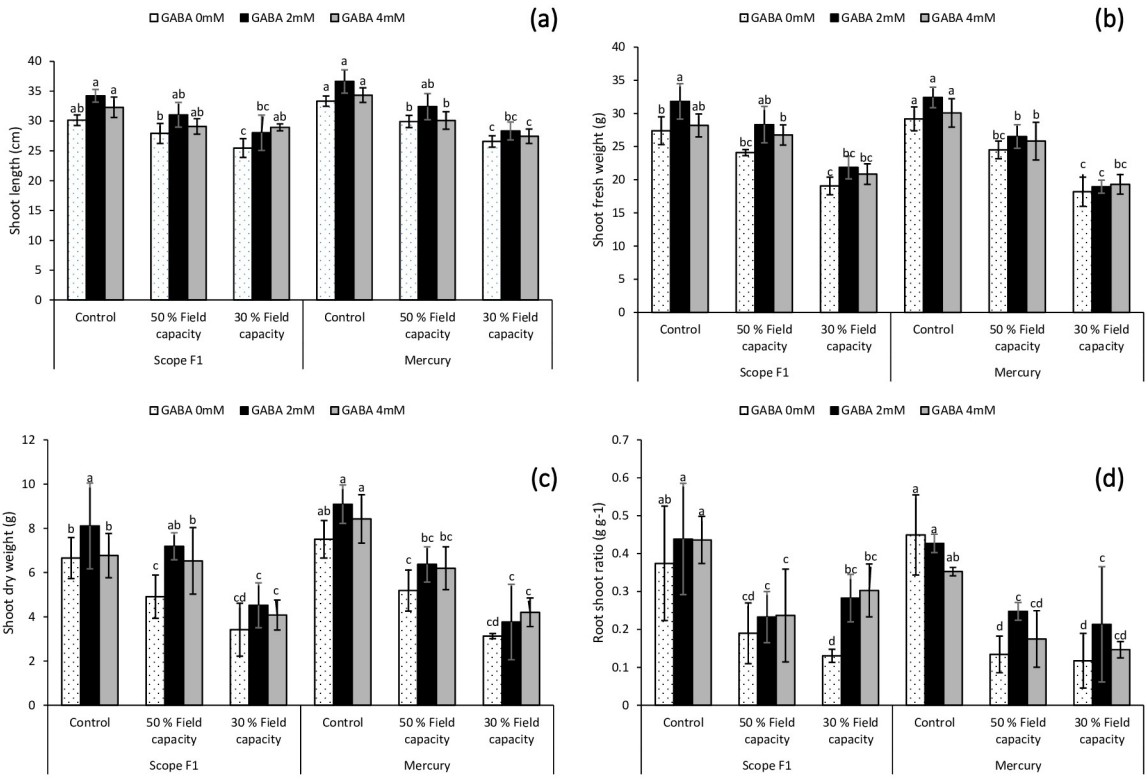

**Fig 2.** Effect of varying levels of exogenously applied GABA on shoot length (**a**) shoot fresh weight (**b**) shoot dry weight (**c**) root shoot ratio (**d**) of sweet pepper (*Capsicum annuum* L.) grown under varying drought stress conditions. Letters (A–D) indicating least significance difference among the mean values at $p \leq 0.05$.

### 3.3. Impact on Chlorophyll "a" content under drought stress

As revealed in Fig 3a, the fresh leaves taken from *Capsicum annuum* L. plants grown under drought stress (i.e. 50% and 30% field capacity) have significantly ($P \leq 0.05$) lowered level of chlorophyll content at harvest. The foliar spray has significantly augmented the levels of chlorophyll "a" in the fresh leaves of *Capsicum annuum* L. under induced drought stress situations at harvest, regardless of the levels of GABA applied or the cultivars studied in the present research. However, 2mM treatment was more effective in 50% field capacity drought stress whereas 4mM GABA showed better result in 30% field capacity drought stress.

### 3.4. Impact on chlorophyll "b" content under drought stress

Fig 3b represented that drought stress significantly declined chlorophyll "b" content in the leaves. Foliar application with GABA solution significantly ($P \leq 0.05$) enhanced chlorophyll "b" content of the fresh leaves of *Capsicum* plants. However, 2mM GABA was effective in 50% field capacity drought stress whereas 4mM GABA showed more enhanced results under 30% field capacity drought stress.

### 3.5. Impact on chlorophyll "a/b" ratio content under drought stress

A change can be seen in the pattern of Chlorophyll a/b ratio content shown in Fig 3c. It was observed that chlorophyll "ab" ratio level was highest in variety Scope F1 and Mercury cultivar under 50% and 30% field capacity drought showing an upsurge in the chlorophyll a/b ratio

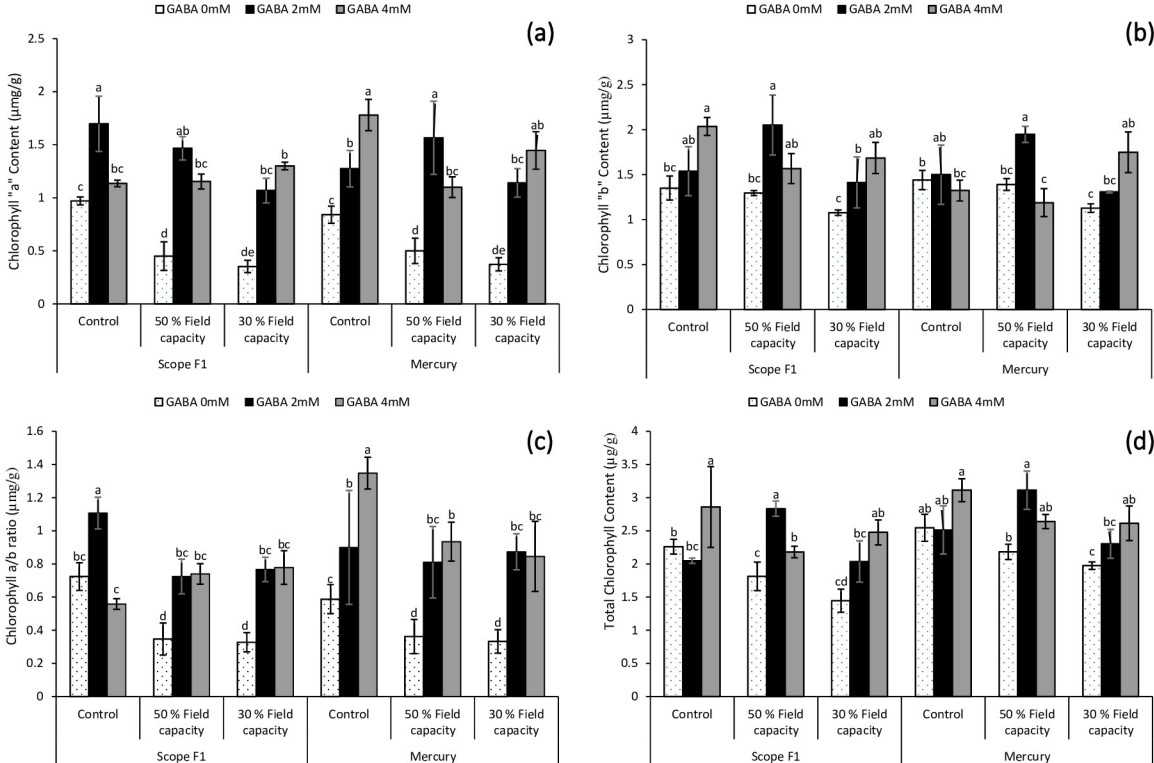

**Fig 3.** Effect of varying levels of exogenously applied GABA on chlorophyll "a" content (**a**) chlorophyll "b" content (**b**) chlorophyll a/b ratio (**c**) total chlorophyll content (**d**) of sweet pepper (*Capsicum annuum* L.) grown under varying drought stress conditions. Letters (A–E) indicating least significance difference among the mean values at $p \leq 0.05$.

under GABA treatment of both, 2mM and 4mM concentrations. This indicated that foliar spraying of growth regulators increases the plant's ability to overcome harsh water stress conditions.

### 3.6. Impact on Total chlorophyll content (TCC) under drought stress

Fresh leaves of *Capsicum annuum* L, were significantly ($P\leq0.05$) affected by the water deficiency stress results in lowering total chlorophyll content. However, the application of GABA at 2mM and 4mM levels showed significant ($P\leq0.05$) increase in the total chlorophyll content (TCC) in leaves irrespective of the cultivar. The GABA application of 2mM showed comparatively better results under 50% field capacity drought stress whereas 4mM GABA was more effective under controlled and 30% field capacity drought stress in both the varieties (Fig 3d).

### 3.7. Impact on Carotenoids content (CC) under drought stress

Significant differences were reported among different amount of drought stress treatments in both the cultivars. The application of GABA spray significantly ($P\leq0.05$) enhanced the Carotenoids content in the plants foliar material (Fig 4a). Carotenoids content in Scope F1 was highest under the 2mM GABA treatment in controlled condition and 50% field capacity drought induced stress. Whereas under 30% field capacity drought 4mM GABA treatment showed highest carotenoids content. Similarly, in Mercury cultivar the carotenoids content showed significance in controlled group and 50% field capacity drought stress treated with

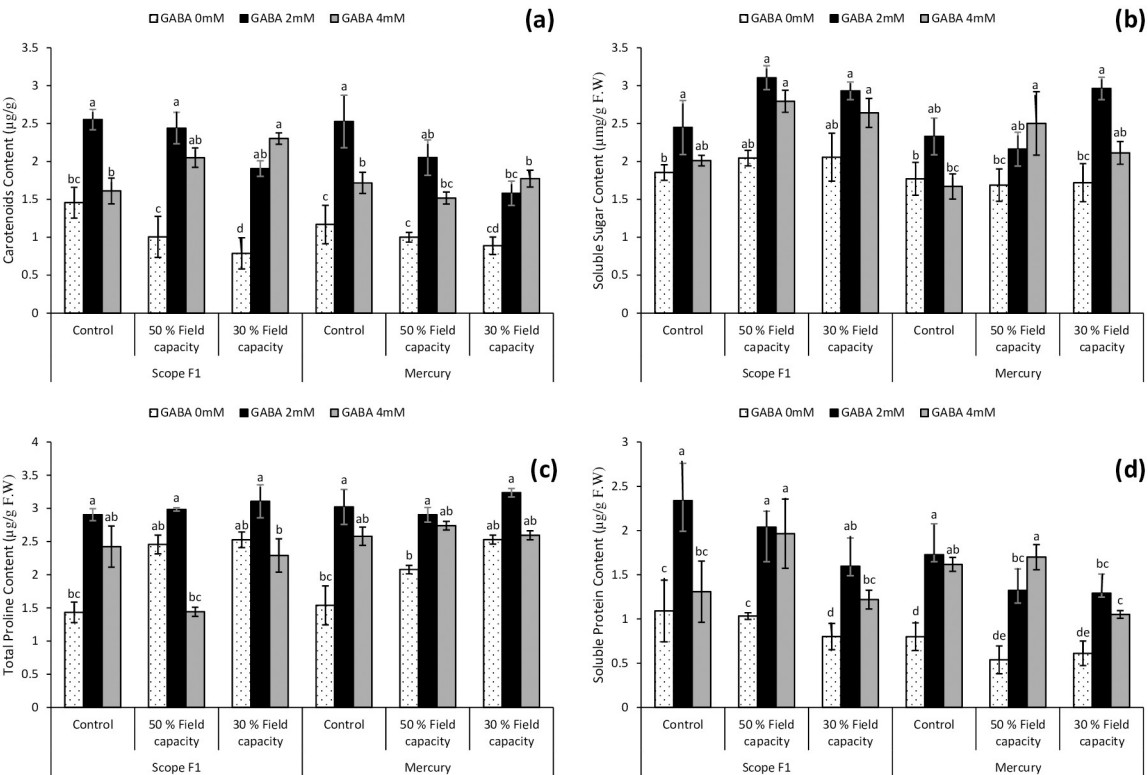

**Fig 4.** Effect of varying levels of exogenously applied GABA on carotenoids content (**a**) soluble sugar content (**b**) total proline content (**c**) soluble protein content (**d**) of sweet pepper (*Capsicum annuum* L.) grown under varying drought stress conditions. Letters (A–E) indicating least significance difference among the mean values at $p\leq0.05$.

2mM-GABA solution, while 4mM-GABA treatment showed highest carotenoids content under 30% field capacity drought plants of the same variety.

### 3.8. Impact on Soluble sugar content (SSC) under drought stress

Growing *Capsicum annuum* L. under drought stress (50% and 30% field capacity) conditions resulted in upsurge of soluble sugar content in the leaves of both the cultivars as compared to controlled group in the present study. Foliar application of 2mM and 4mM GABA significantly ($P \geq 0.05$) amplified the levels of sugar content in the leaves of both the cultivars under both the induced water stress conditions. This indicated that GABA treatments specifically 2mM proliferated the plant's ability to overcome harsh stress conditions and this inherited potential was increased by synthesizing osmoprotectants more rapidly and efficiently when the species is exposed to drought stress. Resultantly, both the varieties showed tolerance with GABA application under induced drought stress regimes (Fig 4b).

### 3.9. Impact on Total proline content (TPC) under drought stress

The fresh leaves of *Capsicum annuum* L. showed an increase in the total proline content when exposed to drought stress in both the studied genotypes. Proline content was further enhanced when the plants were treated with foliar spray of GABA 2mM and 4mM respectively (Fig 4c). However foliar application of 2mM GABA showed better results in increasing the proline content of both the varieties of *Capsicum annuum* L. as compared to 4mM GABA treatments under induced drought stress conditions.

### 3.10. Impact on soluble protein content (SPC) under drought stress

The fresh leaves from sweet pepper plants grown under drought stress conditions exhibited significantly lower levels of soluble protein content irrespective of the cultivars used. Application of 2mM and 4mM Foliar application of GABA solution significantly improved the levels of soluble protein content in the leaves of *Capsicum annuum* L. grown under drought stress of 50% and 30% field capacity in both the cultivars (Fig 4d).

### 3.11. Impact on Peroxidase content (POD) under drought stress

Peroxidase is vital antioxidant defensive enzyme. As shown in Fig 5a, the peroxidase content in the leaves of *Capsicum annuum* L. got increased ($P \leq 0.05$) under drought stress condition in both the varieties. Foliar spray of GABA solution at different levels (2mM and 4mM) has significantly enhanced the peroxidase content in the leaves of sweet pepper irrespective of the genotypes investigated. However, 2mM GABA showed slightly better results in comparison with 4mM GABA treatment under both the water deficient conditions.

### 3.12. Impact on Superoxide dismutase content (SOD) under drought stress

Superoxide dismutase is another important antioxidant defense enzyme that protects plants from the harmful effects of drought stress. The conversion of $O_2$ to $H_2O_2$ and the breaking of $H_2O_2$ is increased during drought stress, resulting in an increase in SOD concentration. Application of foliar spray of 2mM and 4mM GABA significantly increased the Superoxide dismutase content in the fresh leaves of sweet pepper under drought stress in both the cultivars. However, 2mM GABA showed considerable improvement as compared to 4mM GABA under both the conditions of drought stress (Fig 5b).

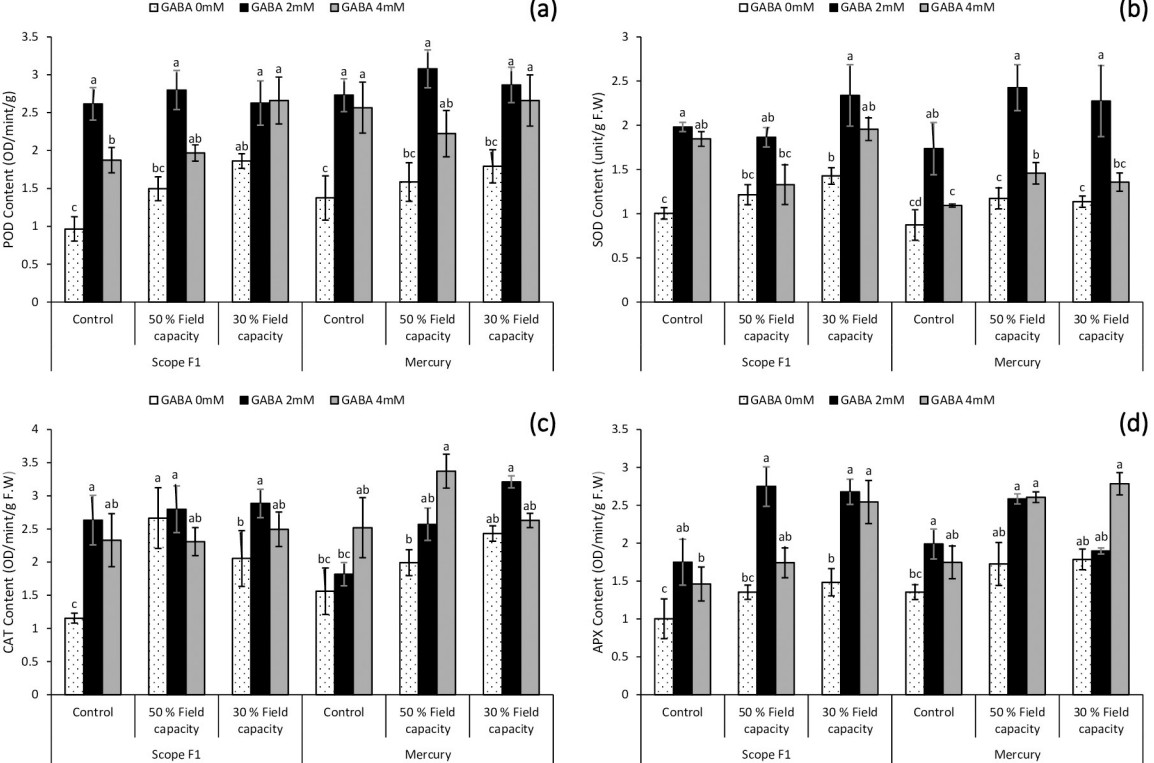

**Fig 5.** Effect of varying levels of exogenously applied GABA on peroxidase content (**a**) superoxide dismutase content (**b**) catalase content (**c**) ascorbate peroxidase content (**d**) of sweet pepper (*Capsicum annuum* L.) grown under varying drought stress conditions. Letters (A–D) indicating least significance difference among the mean values at $p \leq 0.05$.

### 3.13. Impact on Catalase content (CAT) under drought stress

As shown in Fig 5c, leaves of *Capsicum annuum* L. showed a significant ($P \leq 0.05$) increase in catalase activity under both the induced stress conditions in Scope F1 and Mercury variety. However, Foliar application enhanced the activity of CAT and 2mM showed better results in Scope F1 cultivar, Whereas in Mercury cultivar 4mM foliar treatment of GABA solution showed improvement under controlled group and 50% field and 2mM GABA treatment was more effective in 30% field capacity drought induced stress.

### 3.14. Impact on Ascorbate peroxidase content (APX) under drought stress

Results in Fig 5d also indicated that APX activity significantly ($P \leq 0.05$) amplified under drought stress as a defense mechanism to protect plant from ROS formation. Foliar spray treatment also exhibited positive effect on APX activity in both the cultivars by inducing tolerance to drought stress due to high antioxidant defense mechanism by reducing ROS formation. However, foliar spray of 2mM GABA was more effective in Scope F1 variety as compared to Mercury variety where 4mM GABA showed significant results regardless of the drought exposure.

### 3.15. Analysis of multi-correlated biochemical and physiological parameters in *Capsicum annuum* L. treated with foliar spray

Correlation analysis showed in Table 1, estimated positive correlation between Chlorophyll "a," Chlorophyll "b", Chlorophyll a/b ratio, TCC, CC, SPC and POD. Likewise, an optimistic

**Table 1. Analysis of correlation between physiological and biochemical components in *Capsicum annuum* L.**

| Traits | Chl "a" | Chl "b" | Chl a/b | TCC | CC | SSC | TPC | SPC | POD | SOD | CAT | APX |
|---|---|---|---|---|---|---|---|---|---|---|---|---|
| Chl "a" | 1 | | | | | | | | | | | |
| Chl "b" | .578* | 1 | | | | | | | | | | |
| Chl a/b | .897** | 0.172 | 1 | | | | | | | | | |
| TCC | .717** | .635** | .570* | 1 | | | | | | | | |
| CC | .835** | .614** | .676** | 0.453 | 1 | | | | | | | |
| SSC | 0.355 | 0.293 | 0.269 | -0.001 | .613** | 1 | | | | | | |
| TPC | 0.317 | 0.110 | 0.305 | 0.125 | 0.343 | 0.433 | 1 | | | | | |
| SPC | .736** | 0.356 | .701** | 0.287 | .826** | .648** | 0.338 | 1 | | | | |
| POD | .703** | 0.412 | .598** | 0.430 | .701** | .586* | .780** | .544* | 1 | | | |
| SOD | 0.443 | 0.453 | 0.274 | 0.181 | .584* | .685** | .711** | 0.447 | .770** | 1 | | |
| CAT | 0.277 | 0.064 | 0.292 | 0.135 | 0.206 | .571* | .692** | 0.382 | .631** | .538* | 1 | |
| APX | 0.466 | 0.400 | 0.331 | 0.386 | .515* | .569* | .574* | 0.333 | .778** | .567* | .642** | 1 |

Chlorophyll a = Chl "a", Chlorophyll b = Chl "b", Chlorophyll a/b ratio = Chl a/b, Total chlorophyll = TCC, Carotenoids = CC, Soluble sugar content = SSC, Soluble protein content = SPC, Total proline content = TPC, Peroxidase = POD, Superoxide dismutase = SOD, Catalase = CAT, Ascorbate peroxidase = APX,

** correlation is significant at the 0.01 (2-tailed).

* Correlation is significant at the 0.05.

correlation was witnessed between Chlorophyll "b", TCC and CC. All correlations were significant at *p* = .01 and 0.05. In the same way, an optimistic correlation was detected between Chlorophyll a/b ratio, TCC, CC, SPC, and POD, whereas CC showed positive correlation with SSC, SPC, POD, SOD, and APX. Correspondingly, there is positively correlated between SSC, SPC, POD, SOD, CAT and APX. Similarly, TPC evaluated a positive correlation with POD, SOD, CAT and APX. Likewise, SPC showed correlation with POD and POD showed a positive correlation with SOD, CAT, and APX. Correspondingly, SOD was correlated to CAT, and APX, whereas CAT is significantly correlated to APX.

## 3.16. Regression and correlation study of biochemical and physiological parameters in sweet pepper

Study of correlation and regression represents the significant and positive relation between chlorophyll "a" of drought stress and Chlorophyll "a" of GABA treated plants which was r = 0.320 and r = 0.702, chlorophyll "b" of drought stress was r = 0.198 and GABA treated was recorded r = 0.446, whereas similar results were determined for Chlorophyll a/b ratio, TCC, CC, SSC, SPC, CAT, APX. These results clearly demonstrated that GABA treatment significantly increased the biochemical content of *Capsicum annuum* L. The study found that foliar spray had a substantial contribution to the Chlorophyll "a", "b", a/b ratio, TCC, CC, SSC, SPC, of sweet pepper plant. Antioxidant enzymes such as SOD and POD in *Capsicum annuum* L also increased, when compared to water deficit stress treatments. Whereas results recorded for TPC were not significant (Table 2).

## 3.17. The measured "Analysis of variance" of *Capsicum annum* L. traits under drought stress

The results of analysis of variance of the measured attributes revealed substantial variances across genotypes as well as a considerable discrepancy in the interactions between foliar spray components under induced drought stress, as shown in the table below (Table 3).

**Table 2. Regression and Correlation analysis between physiological and biochemical features in *Capsicum annuum* L.**

Regression and Correlation analysis between drought stress and *Capsicum annum* L.

| Chl "a" | Chl "b" | Chl a/b | TCC | CC | SSC | SPC | TPC | POD | SOD | CAT | APX |
|---|---|---|---|---|---|---|---|---|---|---|---|
| $r = 0.320$ | $r = 0.198$ | $r = 0.331$ | $r = 0.391^{***}$ | $r = 0.221$ | $r = 0.349$ | $r = 0.321$ | $r = 0.301$ | $r = 0.270$ | $r = 0.284$ | $r = 0.471^{***}$ | $r = 0.484^{***}$ |
| $R^2 = 0.102$ | $R^2 = 0.039$ | $R^2 = 0.109$ | $R^2 = 0.153$ | $R^2 = 0.049$ | $R^2 = 0.122$ | $R^2 = 0.103$ | $R^2 = 0.091$ | $R^2 = 0.073$ | $R^2 = 0.081$ | $R^2 = 0.222$ | $R^2 = 0.235$ |

Regression and Correlation analysis between GABA and *Capsicum annuum* L.

| Chl "a" | Chl "b" | Chl a/b | TCC | CC | SSC | SPC | TPC | POD | SOD | CAT | APX |
|---|---|---|---|---|---|---|---|---|---|---|---|
| $r = 0.702^{***}$ | $r = 0.446^{***}$ | $r = 0.641^{***}$ | $r = 0.576^{***}$ | $r = 0.576^{***}$ | $r = 0.388^{***}$ | $r = 0.554^{***}$ | $r = 0.191$ | $r = 0.561^{***}$ | $r = 0.320^{***}$ | $r = 0.485^{***}$ | $r = 0.523^{***}$ |
| $R^2 = 0.492$ | $R^2 = 0.199$ | $R^2 = 0.411$ | $R^2 = 0.332$ | $R^2 = 0.331$ | $R^2 = 0.150$ | $R^2 = 0.307$ | $R^2 = 0.036$ | $R^2 = 0.315$ | $R^2 = 0.102$ | $R^2 = 0.235$ | $R^2 = 0.273$ |

Chlorophyll a = Chl "a", Chlorophyll b = Chl "b", Chlorophyll a/b ratio = Chl a/b, Total chlorophyll content = TCC, Carotenoids content = CC, Soluble sugar content = SSC, Soluble protein content = SPC, Total proline content = TPC, Peroxidase = POD, Superoxide dismutase = SOD, Catalase = CAT, Ascorbate peroxidase = APX,

* is significant up to $p = 0.05$, Correlation analysis

*** is significant up to $p$: 0.001.

## 3.18. Analysis of principal components of the biological traits in *Capsicum annuum* L.

The results of Principal component analysis of the biological traits were based on 12 traits and are represented below (Table 4, Fig 6). The first "PC1" accounted for 53.715% of overall variance, which shows a significant correlation with Chlorophyll "a", Chlorophyll a/b ratio, CC, SPC, POD, SOD, and APX predominantly linked with corresponded antioxidant enzymes and osmoprotectants. However, second "PC2" explained 17.047% of complete variance and correlated with TCC, TPC, and CAT, growth response parameters. Similarly, "PC3" explained 9.191% of overall variations with significant variables, e.g., Chlorophyll "b" and SPC. Likewise, "PC4" explained 6.721% of variation with significant variable "SSC".

## 4. Discussion

Many complex mechanisms have been triggered by plants to overcome abiotic and biotic stress. A variety of protective as well as defensive physiological mechanisms play a vital role in minimizing the deleterious effects of stress in plants, including the synthesis of cryoprotectant molecules, an increase in tissue-specific scavenging capacity against excessive ROSs, a decrease in malondialdehyde content, and the production of low molecular weight nitrogenous compounds. GABA is one such chemical that accumulates in plants in response to stress and gives tolerance [46,47]. Numerous studies have shown that exogenous GABA administration significantly raises endogenous GABA levels, most likely via enhancing the activity of tissue-specific GABA transporters [48]. It has been shown that GABA helps plant in tolerance to negative effects of abiotic stress through enhancing antioxidant metabolism as well as adjusting of osmosis and improving or maintaining membrane integrity and hindering the buildup of malondialdehyde (MDA) during lipid peroxidation [49,50].

Scarcity of water has a dramatic influence on the growth characteristics of sweet pepper, as demonstrated by our research. Plant growth characteristics such as roots length, fresh and dry weight as well as shoot length, fresh and dry weight and RSR were all lowered by induced drought stress conditions (Figs 1 and 2). GABA foliar spray was used in the current study at varying concentrations (2mM and 4mM). Regardless of the dose of GABA used in the current study, foliar application of GABA increased the vegetative development of sweet pepper grown under drought-stressed regimes.

**Table 3. The measured "Analysis of variance" of *Capsicum annum* L. traits under drought stress.**

| Traits | Source | SS | df | MS | F | Sig. |
|---|---|---|---|---|---|---|
| | Drought | 0.362 | 2 | 0.181 | 4.993 | 0.035* |
| Chl "a" | Treatment | 2.333 | 2 | 1.167 | 32.174 | 0.000*** |
| | Drought × Treatment | 0.295 | 4 | 0.074 | 2.032 | 0.173 |
| | Error | 0.326 | 9 | 0.036 | - | - |
| | Drought | 0.107 | 2 | 0.053 | 1.377 | 0.301 |
| Chl "b" | Treatment | 0.436 | 2 | 0.218 | 5.619 | 0.026* |
| | Drought × Treatment | 0.572 | 4 | 0.143 | 3.686 | 0.048* |
| | Error | 0.349 | 9 | 0.039 | - | - |
| | Drought | 0.189 | 2 | 0.094 | 2.281 | 0.158 |
| Chl a/b | Treatment | 0.700 | 2 | 0.350 | 8.453 | 0.009*** |
| | Drought × Treatment | 0.027 | 4 | 0.007 | 0.162 | 0.952 |
| | Error | 0.373 | 9 | 0.041 | - | - |
| | Drought | 0.565 | 2 | 0.282 | 4.410 | 0.046* |
| TCC | Treatment | 1.189 | 2 | 0.594 | 9.278 | 0.007*** |
| | Drought × Treatment | 1.039 | 4 | 0.260 | 4.053 | 0.038* |
| | Error | 0.577 | 9 | 0.064 | - | - |
| | Drought | 0.269 | 2 | 0.134 | 2.610 | 0.128 |
| CC | Treatment | 3.984 | 2 | 1.992 | 38.671 | 0.000*** |
| | Drought × Treatment | 0.758 | 4 | 0.189 | 3.676 | 0.049* |
| | Error | 0.464 | 9 | 0.052 | - | - |
| | Drought | 0.577 | 2 | 0.289 | 3.181 | 0.090 |
| SSC | Treatment | 1.931 | 2 | 0.965 | 10.638 | 0.004*** |
| | Drought × Treatment | 0.417 | 4 | 0.104 | 1.148 | 0.394 |
| | Error | 0.817 | 9 | 0.091 | - | - |
| TPC | Drought | 0.500 | 2 | 0.250 | 2.262 | 0.160 |
| | Treatment | 2.795 | 2 | 1.398 | 12.645 | 0.002*** |
| | Drought × Treatment | 0.938 | 4 | 0.235 | 2.122 | 0.160 |
| | Error | 0.995 | 9 | 0.111 | - | - |
| SPC | Drought | 0.529 | 2 | 0.265 | 3.102 | 0.094 |
| | Treatment | 2.645 | 2 | 1.322 | 15.497 | 0.001*** |
| | Drought × Treatment | 0.367 | 4 | 0.092 | 1.074 | 0.424 |
| | Error | 0.768 | 9 | 0.085 | - | - |
| POD | Drought | 0.459 | 2 | 0.230 | 4.719 | 0.040* |
| | Treatment | 4.989 | 2 | 2.494 | 51.244 | 0.000*** |
| | Drought × Treatment | 0.402 | 4 | 0.101 | 2.066 | 0.168 |
| | Error | 0.438 | 9 | 0.049 | - | - |
| SOD | Drought | 0.320 | 2 | 0.160 | 2.020 | 0.188 |
| | Treatment | 2.845 | 2 | 1.423 | 17.991 | 0.001*** |
| | Drought × Treatment | 0.087 | 4 | 0.022 | 0.276 | 0.886 |
| | Error | 0.712 | 9 | 0.079 | - | - |
| CAT | Drought | 1.510 | 2 | 0.755 | 4.919 | 0.036* |
| | Treatment | 1.718 | 2 | 0.859 | 5.597 | 0.026* |
| | Drought × Treatment | 0.501 | 4 | 0.125 | 0.817 | 0.546 |
| | Error | 1.381 | 9 | 0.153 | - | - |
| APX | Drought | 1.504 | 2 | 0.752 | 6.996 | 0.015** |
| | Treatment | 2.356 | 2 | 1.178 | 10.961 | 0.004*** |
| | Drought × Treatment | 0.488 | 4 | 0.122 | 1.135 | 0.399 |

(*Continued*)

**Table 3.** (Continued)

| Traits | Source | SS | df | MS | F | Sig. |
|--------|--------|-----|-----|-----|-----|------|
| | Error | 0.967 | 9 | 0.107 | - | - |

Chlorophyll a = Chl "a", Chlorophyll b = Chl "b", Chlorophyll a/b ratio = Chl a/b, Total chlorophyll = TCC, Carotenoids content = CC, Soluble sugar content = SSC,

Soluble protein content = SPC, Total proline content = TPC, Peroxidase = POD, Superoxide dismutase = SOD, Catalase = CAT, Ascorbate peroxidase = APX,

* is significant up to $p = 0.05$,

** is significant to $p = 0.01$,

*** is significant up to $p = 0.001$.

As an outcome of the collected results, the exogenous administration of GABA boosted all growth factors (Figs 1 and 2). The current study's findings were consistent with those of [51], who discovered that using GABA mitigated the negative impacts of water deficiency stress on developmental constraints of *Piper nigrum* L. (black pepper). These outcomes were also in approval with those currently reported in muskmelon [19], *Oryza sativa* [52], *Trifolium repens* L. [53] *Triticum aestivum* L. [20] and *Vigna radiata* L. [21] grown under saline soil. The exogenous GABA application boosted the fresh plant matter by stimulating cell division and/or expansion, most likely by preserving metabolic balance at the tissue level, which ultimately enhanced the dry weight of root and shoot. GABA's enhancement may be connected to decreased lipid peroxidation and ROS-related damage caused by increased antioxidant activity.

The RWC of plant tissues is a measure used to evaluate metabolic activity [54,55]. Drought stress conditions significantly decreased RWC in the present study (Fig 1d). However, the usage of GABA treatments, particularly 2mM solution lead to an upsurge in RWC. The results of the analysis of GABA's effect on *Capsicum annuum* L. under water scarcity stress exhibited that the optimistic impact of GABA in the studied traits can be credited to refining the GABA effectiveness in water absorption, which can eventually increase the turgor pressure and

**Table 4. Eigen values, individual variance (%), cumulative variance (%), and coefficients of determination of the first three principal components based on biological component correlation matrix.**

| Components | Variance % | | | Components | | | |
|------------|------------|--------------|--------------|------|------|------|------|
| | Eigen Values | Individual % | Cumulative % | PC1 | PC2 | PC3 | PC4 |
| Chl "a" | 6.446 | 53.715 | 53.715 | 0.844 | 0.486 | -0.078 | 0.141 |
| chl "b" | 2.046 | 17.047 | 70.762 | 0.569 | 0.393 | 0.508 | -0.458 |
| Chl a/b | 1.103 | 9.191 | 79.954 | 0.708 | 0.408 | -0.360 | 0.420 |
| TCC | 1.005 | 8.372 | 88.326 | 0.544 | 0.589 | 0.443 | 0.265 |
| CC | 0.537 | 4.479 | 92.804 | 0.857 | 0.299 | -0.182 | -0.275 |
| SSC | 0.359 | 2.990 | 95.794 | 0.698 | -0.373 | -0.260 | -0.428 |
| TPC | 0.213 | 1.772 | 97.566 | 0.659 | -0.546 | 0.095 | 0.270 |
| SPC | 0.128 | 1.071 | 98.636 | 0.768 | 0.183 | -0.514 | -0.151 |
| POD | 0.078 | 0.653 | 99.290 | 0.916 | -0.190 | 0.113 | 0.151 |
| SOD | 0.049 | 0.412 | 99.702 | 0.767 | -0.372 | 0.130 | -0.253 |
| CAT | 0.034 | 0.280 | 99.982 | 0.614 | -0.573 | 0.018 | 0.286 |
| APX | 0.002 | 0.018 | 100.000 | 0.749 | -0.279 | 0.329 | 0.071 |

Chlorophyll a = Chl "a", Chlorophyll b = Chl "b", Chlorophyll a/b ratio = Chl a/b, Total chlorophyll = TCC, Carotenoids content = CC, Soluble sugar content = SSC,

Soluble protein content = SPC, Total proline content = TPC, Peroxidase = POD, Superoxide dismutase = SOD, Catalase = CAT, Ascorbate peroxidase = APX.

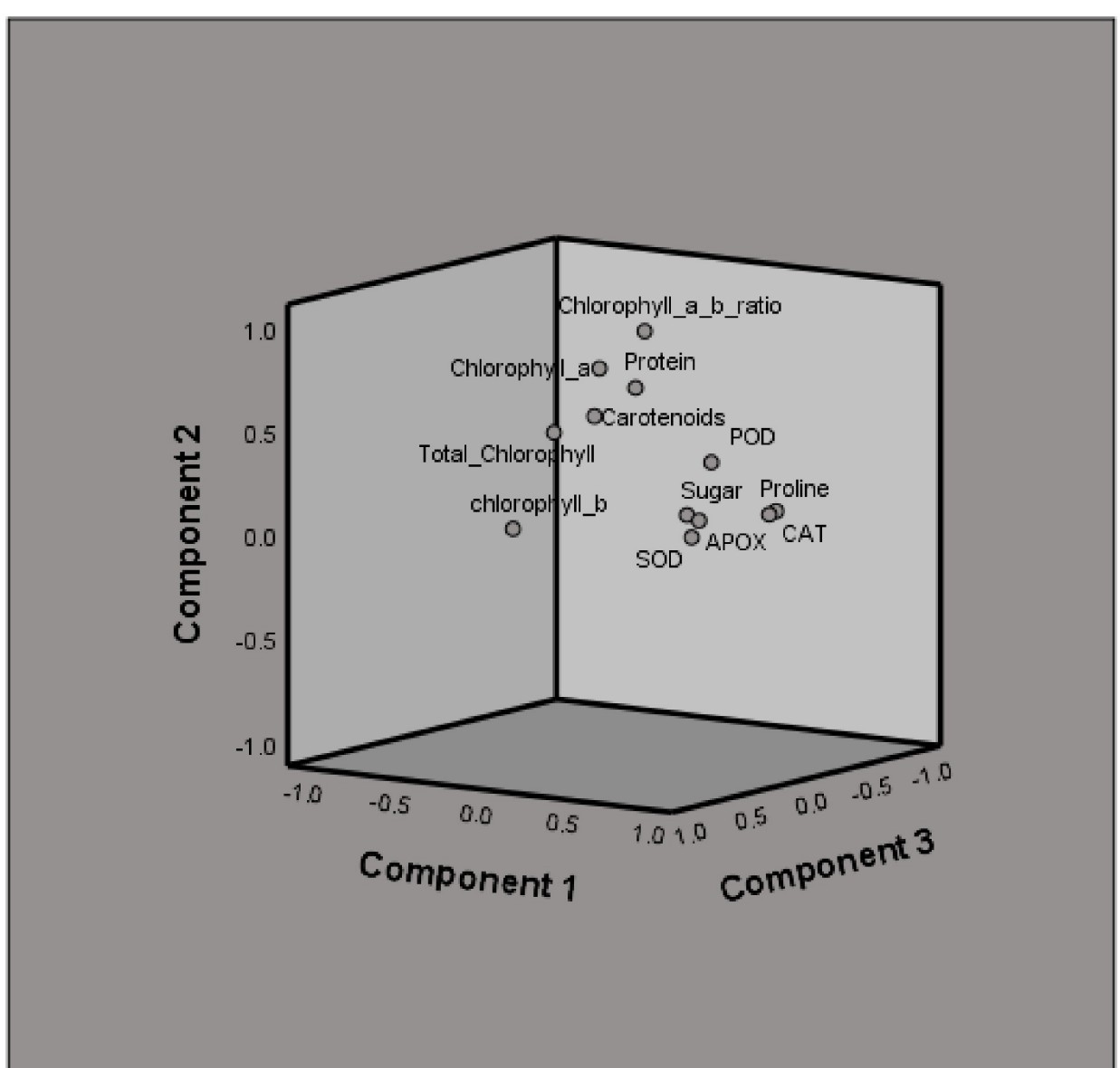

**Fig 6. PC1, PC2, and PC3 loading plots in a rotated space.**

preserve a water balance in the tissues of *Capsicum annuum* L. plant. These findings were parallel with those previously investigated in carrot plant by [56], who discovered that applying GABA mitigated the negative effects of water stress in carrot and improved the RWC of the plant.

The work of [57] showed that GABA improved the level of numerous osmolytes as well as antioxidants in plants, which ultimately help to decrease the level of ROS and MDA. The application of GABA foliar spray enhanced the vegetative development and considerably improved the levels of free plastid pigments, amino acids, as well as the relative water content and antioxidant enzymes in carrots grown under field capacity of 50%. In addition, GABA application also decreased MDA and EL of ions in carrot leaves [56].

GABA reduces constrains of the mitochondrial and photosynthetic activities and the rate of lipid peroxidation to help decrease water scarcity tolerance in *Piper nigrum* L. [51]. Exogenous treatment of GABAs has also been shown to enhance heat tolerance in rice (*Oryza sativa*) [58]. The findings of [59] also showed the significance of GABAs in *Agrostis stolonifera* by boosting its tolerance to drought stress. Research findings have revealed that GABAs alleviates cold damage in many plants such as banana [60], tomato [61], *Cucurbita pepo* [62], *Cucumis sativus*, [63] and peach [64].

Drought stress has negative impacts on root, shoot and leaf's fresh and dry weight of important crops such as *Helianthus annuus* L. [65] and *Brassica napus* L. [66,67], which could be clarified by variations that have taken place at the physio-biochemcal level as a result of water limited condition. Water is being one of the integral part of plant body and act as an important reactant, product, solvent as well as transporter and regulator; besides the transport of nutrients, water plays a crucial part in osmotic adjustment [68]. Additionally, in *Capsicum annuum* L. total biomass reduction may be related to a decrease in turgor pressure [34].

One of the criteria used to calculate the extent of damage done in biological membrane index is electrolyte leakage. The peroxidation of saturated fatty acids in the phospholipid membrane explains why EL occurs during stress. As a result, electrolyte leakage may be computed as a standard for evaluating bio-membrane degradation [69]. The information (Fig 1e) revealed that leakage of electrolytes elevated in reply to increased water deficiency. According to [70], drought stress can cause cell membrane damage in pot marigold plants. The current study found that GABA treatment (2mM and 4mM) decreased the detrimental effects of ROS and boosted plant tolerance to drought stress.

Exogenous GABA administration has also been shown to relieve the detrimental outcomes of drought stress by enhancing membrane integrity of cells in perennial ryegrass [71], muskmelon [19], *Trifolium repens* L. [72], and *Agrostis stolonifera* [59]. These results imply that GABA may have a role in protecting the membrane of sweet pepper leaves cells from the detrimental impacts of water deficit stress, most likely through an anti-oxidative mechanism that led to decrease the level of EL.

Drought is one of the most critical factors affecting *Capsicum annuum* L. growth, inhibiting plant development in its early stages. Drought stress is known to prompt a decrease in respiration, photosynthesis and cause cell membrane injury in higher plants [73]. Furthermore, ROS production as a result of water scarcity lowers chlorophyll as well as carotenoids concentration [74]. Conferring to the findings of this investigation, the decrease in chlorophyll "a" and "b" and carotenoids concentration in *Capsicum annuum* L. might be ascribed to the disastrous impacts of water deficiency stress. Furthermore, the plant's capacity to absorb water is hampered by a lack of available water and salt buildup around the root region causes osmotic stress and ion toxicity, which increases the danger of plant damage [75,76].

The correct estimate of leaf photosynthetic pigments is a crucial component in monitoring plant stress, which varies between plant species and is affected by $CO_2$ in the air, light intensity, soil water content, and temperature [77]. The chlorophyll content and carotenoids content of leaves obtained from the *Capsicum annuum* L. in the present research work showed a prominent decrease when subjected to drought stress. GABA foliar treatment substantially improved the quantity of chlorophyll and carotenoids content in *Capsicum annuum* L. (Figs 3 and 4a). The increased chlorophyll levels in leaves under drought stress caused by GABA administration might be linked to increased cell turgor and the membrane safeguarding the amino acid's function as well as the regulation of countless physio-biochemical reaction within the leaf tissues. Adjusting the osmotic potential of plants by foliar spraying with growth regulator might be regarded a good approach for the upcoming climate change, which would increase drought

tolerance level of important crops in the region. Our findings were consistent with the results of research in a variety of fields and horticultural crops [51,56,59,78–82].

The pattern of change in SSC (Fig 4b) under drought stress was significant in both the varieties, indicating that under stress conditions, an osmoprotective mechanism initiated in the plant for the production of osmoprotectants, such as sugar and proline, to protect the drought exposed plants against the adverse effects of the water stress [4]. The present findings revealed an increased sugar content in the leaves of water-stressed *Capsicum annuum* L. plants subjected to GABA (especially 2mM) application. Under water shortage, proliferation in osmolytes levels such as proline and sugar, is a natural defense mechanism for balancing water potential in plants. Under drought stress, an upsurge in sugar concentration may occur as a way of osmotic potential adjustment [83,84]. The present findings revealed that GABA application in water-stressed *Capsicum annuum* L. triggered an upsurge in the accumulation of soluble sugars. Thus, in the present work, GABA signaling during water-stress is expected to regulate carbohydrate metabolism.

Furthermore, carbohydrates and proline have a significant influence on resistance to water stress in plants. Amino acid such as proline is one of the utmost common amino acids, operating as a well-matched solute for stressful situations [85]. Proline protects plants against abiotic stressors, and its content fluctuates according to the degree of the stress in the plant species [86–88]. As a result, it is not unexpected that proline buildup will rise during water deficit situations. Water stress in *Capsicum annuum* L. triggered the accumulation of the proline and soluble sugars (Fig 4b and 4c). Present evidence are in agreement with earlier reports on the part of GABA in the regulation of osmolytes [89].

GABA application (2mM) is affective in imparting osmotic tolerance in water-stressed plants. GABA application in drought-stressed white clover plants lead to a higher buildup of GABA endogenously which was accompanied by a positive upregulation in proline metabolism and enzymes of GABA-shunt pathway [72,90]. Thus, earlier findings on GABA-induced positive regulation of proline metabolism provided supporting evidence to our present findings. GABA application (2mM) resulted in a surge in proline accumulation in water-stressed *Capsicum annuum* L (Fig 4c).

GABA is also known to be an effective regulator of carbon metabolism and modulates the activity of NADP-dependent isocitrate dehydrogenase (NADP-ICDH) and phosphoenolpyruvate carboxylase (PEPCase) [90,91]. Upregulation of GABA shunt has been attributed to GABA-induced accumulation of TCA cycle intermediates [92,93]. Furthermore, whereas *Thymus vulgaris* L., growth reduced with cumulative water shortage stress, proline and thymol buildup considerably augmented [94,95], which is consistent with the results of our research for *Capsicum annuum* L.

Similarly, water deficit stress reduces the activity of numerous enzymes and causes a net upsurge in free amino acids, which slows down synthesis of protein and/or activation and accelerates protein breakdown in plants [17]. The minimum protein content of Scope F1 and variety Mercury treated with 2mM and 4mM GABA at a level of induced drought of 50% and 30% field capacity at *p*.05 described that this comeback of crop plants rely on the severity and duration of the drought, stage of development, plant species, and the growing season (Fig 4d).

Water deficiency is known to activate both non-enzymatic and enzymatic antioxidative defense mechanism during drought stress which ultimately leads to redox homeostasis [96]. In the current research, our results depicted that drought stress caused a noticeable proliferation in the activity of POD, SOD, CAT, and APX (Fig 5). Under water deficiency stress, there is an increase in proline buildup as well as CAT and APX activities in black cumin [97]. All four enzymes showed a similar pattern, implying that they have an integrative function in regulating water stress tolerance in *Capsicum annuum* L. Exogenous application of GABA positively

upregulated the activity of all the four antioxidant enzymes. These results (Fig 5) indicated that foliar spray treatment showed positive effect on both varieties declaring tolerance to drought stress due to high antioxidant defense mechanism by reducing ROS formation. Altogether, the actions of antioxidative enzymes proliferates in response to reduce the harmful effects of ROS generated in excess during water deficit stress. The activity of all antioxidant enzymes investigated in this study were significantly elevated in the foliar parts of sweet pepper grown under water-stressed circumstances with application of both 2mM and 4mM GABA. Previous reports have established the role of GABA in the modulation of antioxidative defense in *Brassica juncea* L. and *Zea sp*. [89,98].

## 5. Conclusion

Conferring to the findings of the proposed experiment, global climate change would result in severe droughts, which would have a negative impact on ecological fitness and the yield of Pakistan's most valuable crops, including *Capsicum annuum* L. It was founded that foliar spraying methods that are considered to be economically cheap, reasonable, and environmentally acceptable is noteworthy for enzymatic antioxidant activities at the vegetative stage and starting several metabolic processes in the plant, as well as shortening germination time. The agronomic, physiological, and biochemical analyses of the current study showed that foliar GABA application induced changes in plants provide new information that may be useful in enhancing *Capsicum annuum* L. drought tolerance and will lessen the negative effects of the region's impending drought threat due to global variation. More research with other compounds and growth regulators is required in order to distinguish between all foliar treatment techniques in pepper cultivars. In the coming decade, biotechnology and breeding efforts to harness these mechanisms must be increased in order to address the issues of food and fuel security posed by a growing population. Regardless of this, individuals apprehensive with crops development must proceed with prudence and guarantee that potentially harnessed methods are practical in the situation of Pakistan's continuously shifting environment. Additionally, we must acquire appropriate irrigation techniques to lessen the likelihood of an increase in drought in the future, as well as begin the immediate adaptation procedures suggested by the current study and additional related findings of the scholars, in order to maintain our farmland's fertility and boost healthy growth rates in Pakistan.

## Acknowledgments

We are highly acknowledged to the Department of Chemical & Life Sciences, Qurtuba University of Science and Information Technology, Peshawar for providing all facilities regarding this work.

## Author Contributions

**Conceptualization:** Fida Hussain.

**Formal analysis:** Muhammad Saleem khan, Taimur Iqbal, Wadood Shah, Baber Ali, Sezai Ercisli.

**Funding acquisition:** Khalid M. Al Syaad, Sezai Ercisli.

**Software:** Taimur Iqbal, Wadood Shah, Baber Ali, Sezai Ercisli.

**Supervision:** Fida Hussain, Muhammad Saleem khan.

**Writing – original draft:** Babar Iqbal.

**Writing – review & editing:** Wadood Shah, Baber Ali, Khalid M. Al Syaad, Sezai Ercisli.

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
