## [Decision Letter · Decision Letter 0]

13 Jun 2023

PONE-D-23-13341Physiology of gamma-aminobutyric acid treated Capsicum annuum L. (Sweet pepper) under induced drought stressPLOS ONE

Dear Dr. Iqbal,

Thank you for submitting your manuscript to PLOS ONE. After careful consideration, we feel that it has merit but does not fully meet PLOS ONE’s publication criteria as it currently stands. Therefore, we invite you to submit a revised version of the manuscript that addresses the points raised during the review process.

We look forward to receiving your revised manuscript.

Kind regards,

Haitao Shi

Academic Editor

PLOS ONE

Journal Requirements:

5. Please ensure that you refer to Figure 1 in your text as, if accepted, production will need this reference to link the reader to the figure.

6. We note you have included a table to which you do not refer in the text of your manuscript. Please ensure that you refer to Table 1 in your text; if accepted, production will need this reference to link the reader to the Table.

Reviewers' comments:

Reviewer's Responses to Questions

**Comments to the Author**

1. Is the manuscript technically sound, and do the data support the conclusions?

Reviewer #1: Partly

Reviewer #2: Yes

Reviewer #3: Yes

2. Has the statistical analysis been performed appropriately and rigorously? 

Reviewer #1: I Don't Know

Reviewer #2: Yes

Reviewer #3: Yes

3. Have the authors made all data underlying the findings in their manuscript fully available?

Reviewer #1: No

Reviewer #2: Yes

Reviewer #3: Yes

4. Is the manuscript presented in an intelligible fashion and written in standard English?

Reviewer #1: No

Reviewer #2: Yes

Reviewer #3: Yes

5. Review Comments to the Author

Reviewer #1: Though the study is interesting but the current draft needs to be significantly improved. Thus, I suggest major revision.

1. The first 10 lines of the abstract should be concise. The results are not well written in the abstract.

2. Carefully check the entire text and ensure all scientific names are in italics.

3. The introduction is unnecessary is too long. Please significantly shorten it to 1.5 to 2 pages only.

4. In the introduction, for drought and plant growth regulations, authors must add some most recent studies such as doi: 10.1002/tpg2.20279; 10.1016/j.sajb.2023.05.018; 10.1080/15592324.2023.2186045; 10.1016/j.stress.2023.100152; etc.

5. The writing style is not consistent and must be checked and edited.

6. On what basis GABA dose was selected?

7. Change Chloro to Chl throughout the MS. There are many repeated abbreviations. Please define all the abbreviations on the first mention in the main text, then use the abbreviated form.

8. The presentation of results must be carefully improved. For example, authors can add %/folds numerical data for the increased/significant results.

9. So many figures and tables and some figures can be merged into 1 figure and arranged in different legends A/B/C/D etc. For instance, all graphs can be easily arranged in 2-3 figures. Please work on the figures and reduce the total number of figures.

10. The discussion is too long and boring. Please shorten and provide mechanic insights behind the observed changes.

11. Shorten the conclusion.

12. The revised version should be carefully proofread.

Reviewer #2: the paper is very good

the intro is vey long - remove 50% of it

the extensive statistics have to move to supplm. (regression .....)

disscussion need to condense and use updated referances

similariy is high and need to reduce (attached file)

Reviewer #3: Reviewer Comments

As per my analysis, the research paper titled “Physiology of gamma-aminobutyric acid treated Capsicum Annuum L. (sweet pepper) under induced drought stress” submitted by Baber Iqbal is a novel approach toward the field and can be accepted with the following changes.

• Most part of the paper is like an easy, it must be written in scientific manner, better to use indirect form, and also the Use of “WE” is much more than the normal.

• The abstract contains most of the findings in parenthesis, remove these parentheses and make it a plane and easy to understand sentences.

• Add an introductory statement to the abstract and conclusive sentence at the end. Briefly discuss your methods too.

• Introduction and discussion has some irrelevant citations replace them with the following updated relevant works.

• An introduction of almost 2 pages would be sufficient to give an idea of the research done. Some data is given without providing proper citation. If this is not your findings add reference.

• Methodology section is extensively elaborated. it is suggested to make it brief and concise. Each and every method should be cited, if it is not a first time invention.

• Try to merge all the tables in to a single table and add reference values.

• The graphs are properly prepared.

• The results are supported with the present day research which could be further improved with latest citations, it should be discuss thoroughly with existing review.

Important Citations

1. Sami Ullah, Muhammad Ishaq Khan, Muhammad Nauman Khan, Usman Ali, Baber Ali, Rashid Iqbal, Abdel-Rhman Z Gaafar, Bandar M. AlMunqedhi, Sarah Abdul Razak, Alevcan Kaplan, Sezai Ercisli, and Fathia A. Soudy. 2023. Efficacy of Naphthyl Acetic Acid Foliar Spray in Moderating Drought Effects on the Morphological and Physiological Traits of Maize Plants (Zea mays L.). ACS Omega. DOI: 10.1021/acsomega.3c00753.

2. Shah, N.A., Ullah, S., Nafees, M. Exogenous Effect of Sugar Beet Extract On Physio-biochemical Traits of Hordeum vulgare L. Under Induced Salinity Stress. Gesunde Pflanzen (2023). https://doi.org/10.1007/s10343-023-00894-5.

3. Shumaila, Ullah, S. & Nafees, M. Biochar Application to Soil and Seed Pre-Soaking on Growth, Yield and Physiological Response of Solanum melongena L. Under Induced Abiotic Stresses. J Plant Growth Regul (2023). https://doi.org/10.1007/s00344-023-10990-5.

4. Ali, U., Ullah, S. & Nafees, M. 2023. Resistance Induction in Chickpea (Cicer arietinum L.) Against Salinity Stress through Biochar as a Soil Amendment and Salicylic Acid-Induced Signaling. Gesunde Pflanzen. https://doi.org/10.1007/s10343-023-00851-2.

5. Nafees, M., Ullah, S. & Ahmed, I. 2023. Bioprospecting Biochar and Plant Growth Promoting Rhizobacteria for Alleviating Water Deficit Stress in Vicia faba L. Gesunde Pflanzen. https://doi.org/10.1007/s10343-023-00875-8.

6. Fazal Amin, Sami Ullah, Shah Saud, Muhammad,Zahid Ihsan, Shah Hassan, Sunjeet Kumar, Taufiq Nawaz, MatthewTom Harrison, Ke Liu, Imran Khan, Haitao Liu, Khaled El-Kahtany, Shah Fahad. 2023. Hydrothermal time analysis of mung bean (Vigna radiata (L.) Wilczek seed germination at different water potential and temperatures. South African Journal of Botany. 157 (447-456).

7. Gul-Lalay, S. Ullah, M. Nafees, I. Ahmed. 2023. Resistance induction in Brassica napus L. against water deﬁcit stress through application of biochar and plant growth promoting rhizobacteriaBrassica napus L. against water deﬁcit stress through application of biochar and plant growth promoting rhizobacteria. Journal of the Saudi Society of Agricultural Sciences, https://doi.org/10.1016/j.jssas.2023.04.001.

8. Ullah A, Ali I, Noor J, Zeng F, Bawazeer S, Eldin SM, Asghar MA, Javed HH, Saleem K, Ullah S and Ali H (2023) Exogenous g-aminobutyric acid (GABA) mitigated salinity-induced impairments in mungbean plants by regulating their nitrogen metabolism and antioxidant potential. Front. Plant Sci. 13:1081188.

doi: 10.3389/fpls.2022.1081188.

9. Ali, U.; Ullah, S.; 2022. Hydrothermal time model analysis of seed germination responses to osmotic stress and temperatures in Cicer arietinum L. varieties. Journal of Xi’an Shiyou University, Natural Science Edition. 18(12): 1279-1287. http://xisdxjxsu.asia.

10. Lalay, G.; Ullah, A.; Iqbal, N.; Raza, A.; Asghar, M.A.; Ullah, S.; 2022. The alleviation of drought‑induced damage to growth and physio‑biochemical parameters of Brassica napus L. genotypes using an integrated approach of biochar amendment and PGPR application. Environment, Development and Sustainability. https://doi.org/10.1007/s10668-022-02841-2.

11. Raees, N.; Ullah, S.; Nafees, M. 2022. Interactive Eﬀect of Tocopherol, Salicylic Acid and Ascorbic Acid on Agronomic Characters of Two Genotypes of Brassica napus L. Under Induced Drought and Salinity Stresses. Gesunde Pﬂanzen. https://doi.org/10.1007/s10343-022-00808-x.

12. Ullah, A.; Tariq, A.; Zeng, F.; Sardans, J.; Graciano, C.; Ullah, S.; Chai, X.; Zhang, Z.; Keyimu, M.; Asghar, M.A.; et al. Phosphorous Supplementation Alleviates Drought Induced Physio Biochemical Damages in Calligonum mongolicum. Plants 2022, 11, 3054. https://doi.org/10.3390/ plants11223054.

13. Ali, S.; Ullah, S.; Khan, M.N.; Khan, W.M.; Razak, S.A.; Hafeez, A.; Bangash, S.A.K.; Poczai, P. The Effects of Osmosis and Thermo-Priming on Salinity Stress Tolerance in Vigna radiata L. Sustainability. 2022, 14, 12924. https://doi.org/10.3390/su141912924.

14. S. Bibi, S. Ullah, Aqsa Hafeez, M. N. Khan, M. A. Javed, B. Ali, I. U. Din, S. A. K. Bangash, S. Wahab, N. Wahid, F. Zaman, S. K. Alhag, I. H. A. Abd. El-Rahim, A. E. Ahmed and S. Selimm. 2024. Exogenous Ca/Mg quotient reduces the inhibitory effects of PEG induced osmotic stress on Avena sativa L. Brazilian Journal of Biology, 2024, 84, e264642, https://doi.org/10.1590/1519-6984.264642.

15. Arshad, K.; Ullah, A.; Ullah, S.; Bogari, H.A.; Ashour, M.L.; Noor, J.; Amin, F.; Shah, S. Quantifying Osmotic Stress and Temperature Effects on Germination and Seedlings Growth of Fenugreek (Trigonella foenum-graecum L.) via Hydrothermal Time Model. Sustainability 2022, 14, 12049. https://doi.org/10.3390/su141912049.

16. El-Beltagi, H.S.; Sulaiman; Mohamed, M.E.M.; Ullah, S.; Shah, S. Effects of Ascorbic Acid and/or α-tocopherol on Agronomic and Physio-biochemical Traits of Oat (Avena sativa L.) under Drought Condition. Agronomy 2022, 12, 2296. https://doi.org/10.3390/ agronomy12102296.

17. Wadood Shah, Noor Zaman, Sami Ullah and Muhammad Nafees. 2022. Calcium chloride enhances growth and physio-biochemical performance of barley (Hordeum vulgare L.) under drought-induced stress regimes: a future perspective of climate change in the region. Journal of Water and Climate Change; 13(9), 3357 doi: 10.2166/wcc.2022.134.

18. Jalal Khan, Sami Ullah, Sikandar Shah, Sadaf, Sheharyar Khan, Sulaiman. 2022. Modeling the upshots of induced temperature and water stress on germination and seedlings length of radish (Raphanus sativus L.) via hydrothermal time model. Vegetos https://doi.org/10.1007/s42535-022-00490-4.

19. El-Beltagi, H.S.; Shah, S.; Ullah, S.; Sulaiman; Mansour, A.T.; Shalaby, T.A. Impacts of Ascorbic Acid and Alpha-Tocopherol on Chickpea (Cicer arietinum L.) Grown in Water Deﬁcit Regimes for Sustainable Production. Sustainability 2022, 14, 8861. https://doi.org/ 10.3390/su14148861.

20. Saeed, S.; Ullah, A.; Ullah, S.; Noor, J.; Ali, B.; Khan, M.N.; Hashem, M.; Mostafa, Y.S.; Alamri, S. Validating the Impact of Water Potential and Temperature on Seed Germination of Wheat (Triticum aestivum L.) via Hydrothermal Time Model. Life 2022, 12, 983. https://doi.org/10.3390/life12070983.

21. Shakir, U, U.; Ullah, S.; khan, M. N.; Ali, S.; Ali, U., Zaman, A.; Razak, S. A.; Ozdemir F. A. 2022. Effect of Thermopriming and Alpha-Tocopherol Spray in Triticum aestivum L. under Induced Drought Stress: A Future Perspective of Climate Change in the Region. Sains Malaysiana 51(4)(2022): 977-991. http://doi.org/10.17576/jsm-2022-5104-03.

22. Khan, S.; Ullah, A.; Ullah, S.; Saleem, M.H.; Okla, M.K.; Al-Hashimi, A.; Chen, Y.; Ali, S. Quantifying Temperature and Osmotic Stress Impact on Seed Germination Rate and Seedling Growth of Eruca sativa Mill. via Hydrothermal Time Model. Life 2022, 12, 400. https://doi.org/10.3390/ life12030400.

23. Noor, J.; Ullah, A.; Saleem, M.H.; Tariq, A.; Ullah, S.; Waheed, A.; Okla, M.K.; Al-Hashimi, A.; Chen, Y.; Ahmed, Z.; et al. Effect of Jasmonic Acid Foliar Spray on the MorphoPhysiological Mechanism of Salt Stress Tolerance in Two Soybean Varieties (Glycine max L.). Plants 2022, 11, 651. https://doi.org/10.3390/ plants11050651.

24. Ullah, A.; Sadaf, S.; Ullah, S.; Alshaya, H.; Okla, M.K.; Alwasel, Y.A.; Tariq, A. Using Halothermal Time Model to Describe Barley (Hordeumvulgare L.) Seed Germination Response to Water Potential and Temperature. Life 2022, 12, 209. https://doi.org/10.3390/ life12020209.

25. Showler, A. T., Shah, S., Sulaiman., Khan, S., Ullah, S., Degola, F. 2022. Desert Locust Episode in Pakistan, 2018–2021, and the Current Status of Integrated Desert Locust Management. Journal of Integrated Pest Management, (2022) 13(1): 1; 1–11 https://doi.org/10.1093/jipm/pmab036.

26. Nafees, M., Ullah, S., & Ahmed, I. (2022). Modulation of drought adversities in Vicia faba by the application of plant growth promoting rhizobacteria and biochar. Microscopy Research and Technique,1–14. https://doi. org/10.1002/jemt.24047.

6. PLOS authors have the option to publish the peer review history of their article (what does this mean?). If published, this will include your full peer review and any attached files.

Reviewer #1: No

Reviewer #2: **Yes: **Amr Elkelish

Reviewer #3: No

---

## [Author Response · Author response to Decision Letter 0]

27 Jul 2023

Comments by reviewer#1

1 The first 10 lines of the abstract should be concise. The results are not well written in the abstract. 

Answer: The first 10 lines of the abstract have been concised. Results have been also been re-written in a proper manner.

2 Carefully check the entire text and ensure all scientific names are in italics. 

Answer: All the text has been rechecked multiple times for mistakes and all the scientific names are now in italics.

3 The introduction is unnecessary is too long. Please significantly shorten it to 1.5 to 2 pages only. 

Answer: The introduction is shortened to only 2 pages.

4 In the introduction, for drought and plant growth regulations, authors must add some most recent studies such as doi:10.1002/tpg2.20279; 10.1016/j.sajb.2023.05.018; 10.1080/15592324.2023.2186045; 10.1016/j.stress.2023.100152; etc. 

Answer: Some of the most recent studies as mentioned by the respected reviewer have been added into the introduction.

5 The writing style is not consistent and must be checked and edited. 

Answer: The writing style has been checked and edited multiple times and mistakes have been removed.

6 On what basis GABA dose was selected? 

Answer: Based on the findings of the previous research studies GABA doses were applied in the present work. Moreover, prior to pot experiment a Petri dishes lab experiment was conducted, and the same GABA doses were applied. The applied doses proven effective in counteracting drought induced stress and increasing growth and physiological attributes.

7 Change Chloro to Chl throughout the MS. There are many repeated abbreviations. Please define all the abbreviations on the first mention in the main text, then use the abbreviated form. 

Answer: Chloro is changed into Chl throughout the MS. Repeated abbreviations are removed and are now only defined on the first mention in the main text. 

8 The presentation of results must be carefully improved. For example, authors can add %/folds numerical data for the increased/significant results. 

Answer: Results which required improvement have been carefully improved and are now presented in figures.

9 So many figures and tables and some figures can be merged into 1 figure and arranged in different legends A/B/C/D etc. For instance, all graphs can be easily arranged in 2-3 figures. Please work on the figures and reduce the total number of figures. 

Answer: Figures have been merged and arranged in legends as mentioned.

10 The discussion is too long and boring. Please shorten and provide mechanic insights behind the observed changes. Answer: The discussion has been shortened and mechanic insight have been provided behind the observed changes.

11 Shorten the conclusion. 

Answer: The conclusion have been improved and shortened.

12 The revised version should be carefully proofread. 

Answer: Revised version has been carefully proofread.

Comments by reviewer#2

1 the paper is very good the intro is very long - remove 50% of it 

Answer: Thank you for such a great compliment respected sir. Introduction has been shortened and improved.

2 The extensive statistics have to move to supplm. (Regression .....) 

Answer: Statistic have been moved to supplementary (Regression) 

3 discussion need to condense and use updated references 

Answer: Discussion has been condensed and updated references are also added.

4 similarity is high and need to reduce (attached file 

Answer: Similarity has been reduced 

Comments by reviewer#3

1 Most part of the paper is like an easy, it must be written in scientific manner, better to use indirect form, and also the Use of “WE” is much more than the normal. 

Answer: The paper has been re-written in scientific manner and indirect form. The use of “we” is removed.

2 The abstract contains most of the findings in parenthesis, remove these parentheses and make it a plane and easy to understand sentences. 

Answer: Parenthesis in abstract are removed where possible and sentences are made easy to understand.

3 Add an introductory statement to the abstract and conclusive sentence at the end. Briefly discuss your methods too. Answer: Introductory statement and conclusive sentence at the end has been added to the abstract and methods are discussed briefly.

4 Introduction and discussion has some irrelevant citations replace them with the following updated relevant works. Answer: Irrelevant citations have been replaced with updated and relevant work.

5 An introduction of almost 2 pages would be sufficient to give an idea of the research done. Some data is given without providing proper citation. If this is not your findings add reference. 

Answer: Introduction has been reduced to only 2 pages and proper citations are added to the data.

6 Methodology section is extensively elaborated. it is suggested to make it brief and concise. Each and every method should be cited, if it is not a first time invention. 

Answer: Methodology section has been concised and each and every method that is not my work is also cited.

7 Try to merge all the tables in to a single table and add reference values. 

Answer: The first 2 tables have been converted into graphs.

8 The graphs are properly prepared 

Answer: Thank you respected sir.

9 The results are supported with the present day research which could be further improved with latest citations, it should be discuss thoroughly with existing review. 

Answer: The results are further improved with latest citations and are thoroughly discussed with existing review. 

Comments by editor

1 Please ensure that your manuscript meets PLOS ONE's style requirements, including those for file naming. 

Answer: The manuscript has been formatted according to the PLOS ONE style.

2 We suggest you thoroughly copyedit your manuscript for language usage, spelling, and grammar 

Answer: I have thoroughly copyedited my manuscript and language, spelling and grammar mistakes have been removed. 

3 Please ensure that you refer to Figure 1 in your text as, if accepted, production will need this reference to link the reader to the figure. 

Answer: Figures have been updated and now referred to in the text.

4 We note you have included a table to which you do not refer in the text of your manuscript. Please ensure that you refer to Table 1 in your text; if accepted, production will need this reference to link the reader to the Table. 

Answer: I have referred to tables in the correct format and are now in correct number.

---

## [Editor Report · Decision Letter 1]

28 Jul 2023

Physiology of gamma-aminobutyric acid treated Capsicum annuum L. (Sweet pepper) under induced drought stress

PONE-D-23-13341R1

Dear Dr. Iqbal,

We’re pleased to inform you that your manuscript has been judged scientifically suitable for publication and will be formally accepted for publication once it meets all outstanding technical requirements.

Kind regards,

Haitao Shi

Academic Editor

PLOS ONE
---

## [Editor Report · Acceptance letter]

8 Aug 2023

PONE-D-23-13341R1 

Physiology of gamma-aminobutyric acid treated *Capsicum annuum* L. (Sweet pepper) under induced drought stress 

Dear Dr. Iqbal:

I'm pleased to inform you that your manuscript has been deemed suitable for publication in PLOS ONE. Congratulations! Your manuscript is now with our production department. 

Kind regards, 

on behalf of

Dr. Haitao Shi 

Academic Editor

PLOS ONE